# The Many Roads from Alternative Splicing to Cancer: Molecular Mechanisms Involving Driver Genes

**DOI:** 10.3390/cancers16112123

**Published:** 2024-06-01

**Authors:** Francisco Gimeno-Valiente, Gerardo López-Rodas, Josefa Castillo, Luis Franco

**Affiliations:** 1Cancer Evolution and Genome Instability Laboratory, University College London Cancer Institute, London WC1E 6DD, UK; f.gimeno-valiente@ucl.ac.uk; 2Department of Oncology, Institute of Health Research INCLIVA, 46010 Valencia, Spain; gerardo.lopez@uv.es (G.L.-R.); pepa.castillo@uv.es (J.C.); 3Department of Biochemistry and Molecular Biology, Universitat de València, 46010 Valencia, Spain; 4Centro de Investigación Biomédica en Red en Cáncer (CIBERONC), 28029 Madrid, Spain

**Keywords:** cancer driver genes, alternative splicing, oncogenes, tumour suppressor genes, protein variants, oncogenic isoforms, splicing molecular mechanisms, epigenetics, chromatin, splicing factors

## Abstract

**Simple Summary:**

Alternative splicing is a mechanism that allows, through the combination of different exons, for the yielding of several protein variants from a single gene. These variants may display different and often opposed functions. Mutations occurring in driver genes result in oncogenesis, but alternative splicing may also result in obtaining oncogenic variants in the absence of driver mutations. This review describes how the oncogenic potential of driver genes is activated through aberrant alternative splicing. Firstly, there are driver genes directly acting on alternative splicing and their dysregulation affects the splicing of other genes involved in malignant transformation. A second possibility is that aberrant alternative splicing of a proto-oncogene or tumour suppressor gene results in the appearance of an oncogenic variant. The oncogenic potential of a total of 199 driver genes may result from aberrant alternative splicing, and the molecular mechanisms involved are detailed for more than 40 genes.

**Abstract:**

Cancer driver genes are either oncogenes or tumour suppressor genes that are classically activated or inactivated, respectively, by driver mutations. Alternative splicing—which produces various mature mRNAs and, eventually, protein variants from a single gene—may also result in driving neoplastic transformation because of the different and often opposed functions of the variants of driver genes. The present review analyses the different alternative splicing events that result in driving neoplastic transformation, with an emphasis on their molecular mechanisms. To do this, we collected a list of 568 gene drivers of cancer and revised the literature to select those involved in the alternative splicing of other genes as well as those in which its pre-mRNA is subject to alternative splicing, with the result, in both cases, of producing an oncogenic isoform. Thirty-one genes fall into the first category, which includes splicing factors and components of the spliceosome and splicing regulators. In the second category, namely that comprising driver genes in which alternative splicing produces the oncogenic isoform, 168 genes were found. Then, we grouped them according to the molecular mechanisms responsible for alternative splicing yielding oncogenic isoforms, namely, mutations in cis splicing-determining elements, other causes involving non-mutated cis elements, changes in splicing factors, and epigenetic and chromatin-related changes. The data given in the present review substantiate the idea that aberrant splicing may regulate the activation of proto-oncogenes or inactivation of tumour suppressor genes and details on the mechanisms involved are given for more than 40 driver genes.

## 1. Introduction

The “one gene-one enzyme” proposal offered a plausible and fruitful explanation of the molecular basis of heredity at the time it was put forward [1]. Nevertheless, this hypothesis can no longer be accepted in its original formulation. On the one hand, the concept of genes itself is nowadays under discussion [2] and, on the other hand, the existence of alternative splicing (AS) adds further complexity. Anyway, giving up the intricate question of gene definition and considering a gene simply to be a DNA sequence which is transcribed to yield RNA, that is, limiting the gene concept to that of Griffiths and Stotz’s “molecular gene” [3], we have to consider that a single structural gene may produce several mRNAs, which might be translated to give various protein isoforms.

### 1.1. Mechanisms of Alternative Splicing

The existence of splicing to produce mature mRNA was independently reported in 1977 by the groups of Roberts [4] and Sharp [5] working with adenovirus mRNA. Since then, splicing has been shown to occur in most eukaryotic transcripts and in many instances, this process is not limited to the original proposal of intron removal. The molecular mechanisms involved in splicing are complex; they have been the subject of many reviews (see, for instance, [6,7,8,9]) and are summarised in Figure 1A. All these steps require the participation of more than 100 factors, including heterogeneous nuclear ribonucleoproteins (hnRNPs) as well as small nuclear ribonucleoproteins (snRNPs) and enzymes, the latter being dynamically assembled to form intricate complexes termed spliceosomes.

For the purpose of the present review, it must be considered that genes possessing several exons, a common characteristic of most mammalian genes, usually exhibit AS events, which give rise to various mature mRNA isoforms [6]. AS most often results from exon skipping, but other mechanisms contribute to this process too, including alternative 3′ or 5′ splice sites, mutually exclusive exons, and intron retention (Figure 1B). Intron retention has been largely neglected but it deserves a special mention, as it occurs more frequently in cancer than in adjacent normal tissues [10].

The presence of AS suggests a need for the specific selection of introns or exons to be excised during the maturation of pre-mRNA. The mechanisms by which the introns or exons are defined to allow the spliceosome to correctly select them are still poorly understood, although many details are known. The consensus intronic sequence for the 5′ splice site always starts by the dinucleotide GU, which may be followed in humans by A/G, A, G, and U. While the presence of the GU sequence is mandatory, the other nucleotides, especially the last one, occur with less than 20% probability [11]. The consensus intronic sequence for the 3′ splice site in humans is C/U, AG, and it is often preceded by a polypyrimidine tract, although as it occurs in the 5′ splice site (also referred to as donor site), only the dinucleotide immediately next to the exon is always present [12]. Splice sites are often referred to as strong or weak, according to the degree of matching the consensus sequence.

In humans, most splicing events occur co-transcriptionally [13] and the snRNPs U1 and U2AF, carried by the C-terminal domain (CTD) of RNA pol II, are deposited over the 5′ and 3′ splice sites, respectively. The stronger the splice sites, i.e., the tighter the binding of these snRNPs, the easier the assembly of those factors over the splice sites. Nevertheless, it seems obvious that, given the huge number of GU and AG dinucleotides present in the genome and the relatively low probability of occurrence of the rest of the nucleotides in the consensus sequences, further factors are required to assure a proper assembly of the snRNPs and the subsequent initiation of the splicing mechanism. Several cis- and trans-acting elements that regulate assembly of splicing machinery have been described [14]. The cis elements are intronic splicing enhancers (ISEs), exonic splicing enhancers (ESEs), intronic splicing silencers (ISSs), and exonic splicing silencers (ESSs), according to their location and function (Figure 2). The interplay between splice sites and cis regulatory elements results in a complex net of interactions [15]. The organisation of chromatin and its epigenetic modifications [16,17] and the phosphorylation of the C-terminal domain of RNA pol II [18] also have an influence on the regulation of spliceosome assembly.

In the preceding paragraphs, only AS has been considered the mechanism by which a single gene may give rise to various mRNA isoforms. Other mechanisms exist that may produce an isoform multiplicity. For instance, the presence of alternative promoters within a gene body, results in the appearance of isoforms shorter than the canonical one. Finally, the possibility of alternative polyadenylation signals also may produce isoforms of different length (Figure 1B). Although these two mechanisms are not, strictly speaking, AS events, they will be included in the present review because they give rise to alternative isoforms, often with different functional properties.

### 1.2. Alternative Splicing and Disease

Not all the mRNA isoforms resulting from AS or related mechanisms are translated into protein. Often, some the mRNA isoforms incorporate “poison” exons containing premature stop sequences and are degraded by nonsense-mediated decay (NMD), a surveillance mechanism which also eliminates other abnormal mRNA isoforms. The mechanism of NMD is outside the scope of the present review. The interested reader may consult the recent review by Pawlicka et al., which also deals with the role of this process in cancer [19]. Other mRNA isoforms either are present in negligible levels in some tissues or are not translated to protein.

To centre upon the mRNA isoforms that are eventually translated, it must be pointed out that the different resulting protein isoforms may have different or even opposite functions. A classical example, described in 1994, is that of the Fas Cell Surface Death Receptor (*FAS*) gene, which is translated to a mitochondrial membrane protein of the Tumour Necrosis Factor (TNF) receptor superfamily and contains a death domain. The AS of exon 6 results in a soluble isoform lacking the transmembrane domain. While the canonical protein possesses an apoptotic function, the soluble form is anti-apoptotic [20]. After this initial finding, many cases of protein isoforms with different functions have been described. It is not surprising that numerous examples exist in which isoforms derived from aberrant splicing are related to several pathologies. This issue has been extensively studied and it is recognised that many single-nucleotide variants result in pathologic outcomes by causing alterations in AS [21]. Specifically, the influence of AS in cancer has also been the subject of many studies [22,23,24,25].

Aberrant AS may be produced by a wide variety of causes (Figure 3). For instance, there are mutations that affect splicing either in cis or in trans, i.e., by modifying the DNA sequences or the splicing factors. Among the former, mutations in splice sites or in regulatory sequences, either intronic or exonic, may result either in the appearance of novel isoforms or in an aberrant balance of the different splicing variants. Mutations in the core units of the spliceosome or in other splicing factors, as well as alterations in their levels, also affect the results of the splicing process. In some instances, competitive inhibition of the binding of factors to their cis sequences also occurs. Post-translational covalent modifications of splicing factors also change their capacity of producing a normal set of isoforms. AS is also affected by the structure and epigenetic modifications of chromatin and by the post-transcriptional modifications of pre-mRNA.

### 1.3. Cancer Driver Genes

Cancer results from a series of alterations in gene function which cause an uncontrolled cellular growth and, eventually, the dissemination of transformed cells. Classically, those alterations were envisaged as resulting from genetic mutations [26]. Somatic mutations are extremely frequent during development, but most of them are irrelevant to the acquisition of a cancerous phenotype. In 2009, Stratton et al. established a clear-cut distinction between mutations that confer a growing advantage to the cell that carries them and those that do not give it that advantage. The first type of mutation leads to a Darwinian-like superiority in the microenvironment of the tissue in which cancer develops and is called “driver mutations”, while the second class of mutations, referred to as “passenger mutations”, neither confers growing advantages nor is selected [27]. Driver mutations occur in genes controlling the homeostasis of key functions; these genes are referred to as cancer driver genes and their malfunction affects one or more of twelve signalling pathways [28]. Typically, cancer driver genes are oncogenes or tumour suppressor genes. Although their definition is somewhat controversial [29], we adopt here the commonest concept of oncogenes and tumour suppressor genes. In the former case, the driver mutations activate the genes or cause new oncogenic functions. On the contrary, driver mutations are inactivated in suppressor genes. Criteria to identify genes as cancer drivers were established early [30] and the list of driver genes is continuously growing; while some 140 genes were identified as drivers of cancer in 2013 [28], the list of mutational driver genes increased to 568 in 2020 [31].

It is important to note that alterations in the functionality of a gene may be acquired not only due to changes in copy number or to mutations in the gene body or in control regions, but also due to epigenetic modifications. For instance, promoter methylation may inactivate suppressor genes and drive cancer. Epigenetic hits parallel in this way the effects of the classical two-hit hypothesis based on genomic mutations. In this context, a method to differentiate driver DNA methylation changes from passenger events has been reported [32]. Aberrant AS may also be the cause of activating the cancer-driving potential of several genes and the purpose of the present review is to analyse the different AS-related processes which, involving driver genes, result in neoplastic transformation, with an emphasis on their molecular causes.

## 2. Alternative Splicing and Cancer Driver Genes

### 2.1. Alternative Splicing in Cancer

Aberrant AS may lead to the transcription and translation of a mutation-independent, active isoform of an oncogene [33]. For example, AS of the NFE2-Like BZIP Transcription Factor 2 (*NFE2L2*) gene frequently results in the skipping of exon 2 in lung cancers and head and neck cancers, and the outcome of this event is the activation of the proto-oncogene in the absence of known mutations [34]. Aberrant AS may also result in a mutation-dependent isoform of an oncogene with different oncogenic properties or, finally, in a mutation-independent, inactive isoform of a suppressor gene. Recently, Singh et al. [35] have identified critical splicing factors that regulate AS during embryonic development and are reactivated in tumours, causing aberrant splicing. For instance, Splicing Factor 3b Subunit 1 (SF3B1) and U2 Small Nuclear Auxiliary Factor 2 (U2AF2), which will be mentioned below, are among these critical splicing factors.

Cancer driver genes may be associated with AS in two ways: either they control the splicing of other genes with the eventual result of driving the neoplastic transformation, or their pre-mRNA is subject to splicing events that activate their oncogenic potential. We revised the literature to check how many of the 568 genes identified as drivers by Martínez-Jiménez et al. [31] are involved in AS in either of the two ways mentioned above.

### 2.2. Driver Genes That Control the Alternative Splicing of Other Genes

A total of 31 out of the 568 genes included in the list of Martínez-Jiménez et al. [31] control in some way the splicing of other genes, yielding in turn oncogenic isoforms (Table 1). The proteins encoded by the genes in Table 1 mainly fall into three categories: components of the snRNPs constituent of the spliceosomes, splicing factors, and regulators of splicing. The distinction among these two latter classes of proteins is somewhat artificial. We have named splicing factors those that directly interact with pre-mRNA, while we consider splicing regulators the proteins that modulate the splicing patterns in an indirect way, for instance, by interacting with splicing factors or components of the splicing machinery. This criterion has been used to sort the genes in Table 1. It should be noted that some of them play other physiological roles apart from being involved in alternative splicing.

It seems obvious that mutations, changes in expression, or any other cause that alters the concentration of the proteins encoded by components of the spliceosome or general splicing factors result in changes of the splicing patterns of many genes in many cancers. This is the case of genes *SF3B1*, RNA-Binding Motif Protein 10 (*RBM10*), Small Nuclear Auxiliary Factor 1 (*U2AF1*), Serine and Arginine-Rich Splicing Factor 2 (*SRSF2*), Elongation Factor Tu GTP-Binding Domain-Containing 2 (*EFTUD2*), RNA-Binding Motif Protein 39 (*RBM39*), Heterogeneous Nuclear Ribonucleoprotein A2/B1 (*HNRNPA2B1*), RNA-Binding Fox-1 Homolog 2 (*RBFOX2*), *U2AF2*, SUZ12 Polycomb Repressive Complex 2 Subunit (*SUZ12*), RNA-Binding Motif Protein 38 (*RBM38*), Bromodomain Containing 4 (*BRD4*), and Zinc Finger CCCH-Type, RNA-Binding Motif, And Serine/Arginine Rich 2 (*ZRSR2*) (Table 1).

Among these genes, *SF3B1* deserves special attention. It is the most frequently mutated gene among those coding for spliceosome components, especially in myelodysplastic syndromes, chronic lymphocytic leukaemia, chronic myelomonocytic leukaemia, uveal and skin melanoma, and breast and pancreatic cancers (see [36] and references therein). Hotspot mutations of *SF3B1* occur within the C-terminal HEAT domains (residues 622–781) and they are considered neomorphic mutations [36,37]. For instance, mutation K700E, which presumably alters the interaction between *SF3B1* pre-mRNA and the polypyrimidine tract-binding factor U2AF2, results in the selection of an alternative upstream branching point in the target genes, which, in turn, causes the selection of a different 3′ splice site. As a final consequence, the resulting aberrant transcripts may be either translated to an aberrant protein or undergo NMD [36,37]. These results are consistent with the transcriptomic analysis of Wang et al. [38], which showed that SF3B1 mutations in chronic lymphocytic leukaemia result in changes in splicing patterns, with a special incidence of alternative 3′ splice sites. In the instance of the Transcriptional Repressor (*SPEN*) family, alteration in splicing has only been reported in ovarian cancer [39]. The Nuclear Receptor-Binding SET Domain Protein 2 (*NSD2*) gene represents a singular situation, as it encodes two major protein isoforms, namely MMSET II and REIIBP. The latter interacts with the SMN (survival of motor neuron) complex, which, in turn, is involved in the assembly of spliceosomal ribonucleic particles in multiple myeloma cells [40].

**Table 1 cancers-16-02123-t001:** Driver genes that control the splicing of other genes to originate oncogenic isoforms.

Gene	Other Roles	References
Spliceosome components		
*SF3B1*		[37,41]
*U2AF1*		[42,43]
*EFTUD2*		[44,45]
*NSD2*		[40]
*U2AF2*		[35]
*ZRSR2*		[43]
Splicing factors		
*RBM10*		[46]
*SPEN*	transcriptional repressor	[39]
*SRSF2*		[43,47,48,49]
*RBM39*		[50]
*HNRNPA2B1*		[51]
*RBFOX2*		[52,53]
*RBM38*		[54]
*DAZAP1*		[55,56,57,58]
Splicing regulators		
*SOX9*	transcriptional factor	[59,60]
*MTOR*		[61]
*CTCF*		[62]
*RUNX1/RUNX1T1*	transcriptional repressor	[63]
*FUBP1*		[64]
*RANBP2*		[65]
*DROSHA*	component microprocessor	[66,67]
*ARID1A*	component SWI/SNF	[68]
*SETDB1*	histone methyltransferase	[69]
*ZEB1*	transcriptional factor	[70]
*PSIP1*		[71,72]
*HNF1A*	transcriptional factor	[73]
*NONO*	transcriptional factor	[74,75]
*WT1*	transcriptional factor	[76,77,78]
*SUZ12*	component Polycomb complex	[79]
*BRD4*		[80]
*QKI*		[53,81]

Among the genes classified as “splicing regulators”, a variety of mechanisms account for their influence on AS. Firstly, there are genes which regulate the transcription of splicing factors, and this results in an obvious modification of the splicing patterns. Mechanistic Target Of Rapamycin Kinase (*MTOR*), for instance, encodes the serine/threonine kinase mTOR, which controls the transcription of Serine And Arginine-Rich Splicing Factor 3 (*SRSF3*), with the subsequent transcriptome-wide increase in AS events, especially exon skipping [61]. The expression of SRSF3 and other splicing factors, such as SF3B1, is also regulated by the histone methyltransferase encoded by SET Domain Bifurcated Histone Lysine Methyltransferase 1 (*SETDB1*) [69], and the Wilms tumour suppressor zinc-finger transcription factor, encoded by WT1 Transcription Factor (*WT1*), activates the transcription of SRSF Protein Kinase 1 (*SRPK1*) and Serine And Arginine Rich Splicing Factor 1 (*SRSF1*) [76,77]. The fusion oncogene RUNX Family Transcription Factor 1 (RUNX1)/RUNX1 Partner Transcriptional Co-Repressor 1 (*RUNX1T1*) also controls the expression of several splicing factors, among which is the driver gene RNA-Binding Fox-1 Homolog 2 (*RBFOX2*) [63]. Zinc Finger E-Box-Binding Homeobox 1 (ZEB1) is a transcription factor (Table 1) which regulates AS in an indirect manner through repressing the transcription of the gene encoding the epithelial splicing regulatory protein ESRP1. Osada et al. have found that the level of ZEB1 in mesenchymal-like cells from oral squamous-cell carcinoma is high and, consequently, these cells express *ESRP1* to a low level. The change in the concentration of ESPR1 correlates with a switching in the isoforms of the receptors of the fibroblast growth factor, which occurs during the epithelial–mesenchymal transition [70].

A second mechanism of action of the splicing regulators included in Table 1 is the capacity of their products to bind pre-mRNA. For instance, the KH Domain-Containing RNA-Binding (QKI) protein may compete with the splicing factor SF1 for binding to the branchpoint sequence and, in this way, suppress the splicing patterns associated with lung cancer [81]. A more indirect effect of QKI on splicing has been reported in several cancers. It will be commented later that the splicing factor Muscleblind-Like Splicing Regulator 1 (MBNL1) controls an exon-skipping event in the driver gene Nuclear Mitotic Apparatus Protein 1 (*NUMA1*). QKI also controls this event, albeit in an opposite sense, presumably by binding *NUMA1* pre-mRNA [53]. Non-POU domain-containing octamer-binding protein, encoded by the *NONO* gene, is also an RNA-binding protein. Binding of NONO especially influences intron retention, but also exon skipping, usage of alternative 3′ and 5′ splice sites, and mutually exclusive exons. Nevertheless, two different mechanisms have been reported for its role in AS-related cancer development. NONO binds the pre-mRNA of the SET Domain And Mariner Transposase Fusion Gene (*SETMAR*) and interacts with the splicing factor SFPQ to regulate its splicing, favouring the inclusion of *SETMAR* exon 2 to give the long isoform. This isoform, through the SET domain of *SETMAR*, catalyses the methylation of H3 lysine 27 at the promotor of oncogenes involved in metastasis, with its subsequent suppression. In patients with bladder cancer with lymph node metastasis, *NONO* is downregulated, and a low level of NONO has been correlated with the onset of lymphatic metastasis in bladder cancer through that complex molecular mechanism [75]. However, in glioblastoma multiforme, the expression of *NONO* is increased and is associated with poor survival of patients. NONO binding to the single intron of *GPX1* favours splicing of the two exons, while the knockdown of *NONO* results in intron retention and in the inhibition of tumour growth [74]. Finally, SRY-Box Transcription Factor 9 (SOX9) plays an alternative role in the regulation of AS. Also being an RNA-binding protein, it is associated with other RNA-binding splicing factors, such as the core exon junction complex component Y14, and in this way, it affects the splicing of many cancer-related genes [59,60].

In opposition to SOX9, the role played by HNF1 Homeobox A (HNF1A) in AS derives from its being a transcription factor. HNF1A specifically activates the transcription of the proto-oncogene *SRC* from an alternative promoter, giving rise to the long isoform c-SRC, whose transcripts are differentially present in several cancers, depending on the level of HNF1A [73].

The mechanisms of action of the remaining splicing regulators listed in Table 1 display unique features. For instance, the CCCTC-Binding Factor (*CTCF*) gene codes for a well-known and ubiquitously expressed zinc finger-containing protein. Among its diverse roles, CTCF may mediate AS in different ways. The first-reported one derives from its capacity to bind non-methylated CpG-rich sequences. When these sequences occur after an alternative exon, the consequent stalling of RNA polymerase may favour the inclusion of the exon, but, alternatively, it may provide time enough for negative regulators to bind ISS or ESS, resulting in exon skipping [14]. In recent years, several other AS-related functions of the binding of CTCF to DNA have been reported. For instance, CTCF-mediated DNA intragenic looping between promoters and intronic regions upstream an alternative exon may bring into contact this distal exon with the splicing factors localised at the promoter [14], resulting in the inclusion of the alternative exon [82], although it has not been verified whether this mechanism affects cancer-related AS events. The re-organisation of chromatin 3D architecture by CTCF–cohesin binding to distant sequences may also have an influence on alternative transcription termination, when one of the anchoring sites of CTCF–cohesin lies close to one of the several possible polyadenylation sites of a gene. As the binding of CTCF depends on the methylation state of the target sequence, DNA methylation influences alternative polyadenylation in this manner, and this mechanism has been shown to result in aberrant transcriptome diversity in cancer cells [83]. Other putative roles of CTCF in AS, involving modifications in chromatin structure, have been proposed and recently reviewed [62]. In summary, CTCF may favour either exon skipping or inclusion and thus play a multi-faceted role in AS.

The singular role of Far Upstream Element-Binding Protein 1 (FUBP1) in the regulation of AS consists in its capacity of binding the Vir-Like M6A Methyltransferase Associated (VIRMA) and RNA-Binding Motif Protein 15 (RBM15) components of the complex responsible for the methylation of the N6 atom of the purine ring of adenosine to give m6A. FUBP1 binding facilitates the recruitment of the complex to its target sites and the formation of m6A favours the normal usage of downstream splice sites. Consequently, the downregulation of *FUBP*1 results in the use of alternative splice sites. This occurs in many cancer-related genes, some of them being driver genes themselves, such as Caspase 8 (*CASP8*), Tumor Protein P53 (*TP53*), MDM4 Regulator Of P53 (*MDM4*), BRCA1 DNA Repair-Associated (*BRCA1*), and Dicer 1 Ribonuclease III (*DICER1*) [64]. Finally, RAN-Binding Protein 2 (*RANBP2*) is responsible for maintaining the morphology of nuclear speckles, which, in the absence of the factor encoded by the gene, are converted into cytoplasmic granules [52]. It is known that inactive SR proteins are confined in nuclear speckles, where they are bound to the long ncRNA metastasis-associated lung adenocarcinoma transcript 1 (*MALAT-1*). Phosphorylated SR proteins are released from the nuclear speckles when needed for accomplishing their role in splicing [84], but, in the absence of RANBP2, they remain sequestered within the cytoplasmic granules, with the subsequent alteration in the splicing of many genes [65].

AT-Rich Interaction Domain 1A (*ARID1A*), which encodes a subunit of the chromatin remodelling complex BAF—the mammalian SWI/SNF—participates in the regulation of splicing in an indirect manner. The ARID1A protein is involved in the splicing of the Spermidine/Spermine N1-Acetyltransferase 1 (*SAT1*) gene, which encodes a crucial enzyme in the catabolism of polyamines. The *SAT1* gene contains a cryptic exon, called exon X, which includes three stop codons. Its inclusion yields an mRNA which is degraded by an NMD process, with the result of lowering the level of the SAT1 enzyme and of increasing those of spermine and spermidine. As these polyamines are upregulated in cancer cells and favour proliferation [85], alterations in the level of SAT1 have consequences in tumour growth. ARID1A promotes the exclusion of exon X of *SAT1* by facilitating the binding of the histone lysine dimethyl-demethylase KDM3A to the chromatin region encompassing exon X, which, in turn, recruits SRSF3 with the final result of incorporating exon X [68].

### 2.3. Oncogenic Isoforms of Driver Genes Resulting from Alternative Splicing

Apart from the driver genes that modulate the splicing of other genes leading to oncogenicity, which have been dealt with in the preceding section, we found that in 168 genes out of the 568 mentioned by Martínez-Jiménez et al. [31], AS events give rise to oncogenic isoforms. Some of the genes listed in Table 1 are themselves subject to AS. Appendix A displays a list of these 168 genes, together with the relevant AS events and mechanisms involved in the splicing when known. Exon skipping is the most frequent event related to oncogenesis, as it was found in 94 out of the 168 genes in Appendix A. Exon skipping was followed by the occurrence of alternative termination or initiation sites, alternative 5′ and 3′ splicing sites, and intron retention (Figure 4A). These data depart from the observed frequency of AS events in human genes. Actually, although exon skipping is the most common event in humans, intron retention only accounts for 5% of events, and the use of alternative 5′ splice sites, which ranks second in our case, is also scarcely found (7.9%) in human genes as a whole [86].

The alterations in the AS patterns of the genes are originated from different molecular causes. Although, in several instances, the precise cause has not been determined, in 97 out of the 168 genes in Appendix A, the causes are known. The most frequent one, occurring in nearly 50% of AS events, is the mutation of splice sites, branching points, or binding sites of splicing regulatory elements (ESEs, ISEs, ESSs, or ISSs), as shown in Figure 4B. Alterations in the splicing factors or regulators, either by mutation or by up- or downregulation of their synthesis, represent the following cause, accounting for one-third of events. Epigenetic and/or chromatin changes are responsible for a relatively significant number of events (more than 10%). The characteristics of the most relevant genes, either by their significance in oncogenesis or by the peculiarity of their splicing mechanisms, will be dealt with in the next section.

## 3. Some Specific Examples of Alternative Splicing in Driver Genes

### 3.1. Alternative Splicing Caused by Mutations in Cis Splicing-Determining Elements

Mutations in cis splice elements usually result in altered splicing patterns, which may give rise to cancer driver isoforms. Most of these mutations occur in splice sites and the following paragraphs summarise the most relevant driver genes in which these mutations have been described.

*TP53* is a suppressor gene coding for p53, a protein frequently dubbed “the guardian of the genome” because of its role in maintaining genetic stability. Its primary role is that of transcription factor, and its target genes amount to 3661 [87]. It stands in the first place in the compendium of mutational cancer drivers [31] used in the present review due to the high rate of inactivating mutations in many cancer types. The occurrence of AS in human *TP53* was first reported in 1996 [88] and, since then, many articles have dealt with the nature and characteristics of its many isoforms. The gene comprises 11 exons and may be transcribed to several isoforms through different events, which include exon skipping, alternative promoters, intron retention, inclusion of cryptic exons, alternative 3′ splice sites, and alternative termination [89,90,91,92,93]. The isoforms p53β and p53γ, which result from the alternative splicing of exons 9β and 9γ located within TP53 intron 9, are of special interest because they modulate the expression of the canonical isoform and are downregulated in cancer tissues [94].

The IARC TP53 Database (https://www.iarc.who.int/news-events/new-release-of-the-iarc-tp53-database/, accessed on 7 March 2024.), released in 2012, compiled the mutations found in human *TP53* gene in tumour samples, and a systematic analysis of somatic mutation data extracted from this database and from genomic data repositories has been published [95]. Afterwards, the consequences of mutations in splice sites have been especially studied in colorectal cancer [96]. This study covered 401 primary cancer specimens and sequencing revealed the existence of some mutations not included in the IARC database.

Mutations in *TP53* were detected in 241 samples, most of them localised in splice sites of exons 5–8, which encode the DNA-binding domain. Mutations in splice sites were found in 16 tumours and were located at exon 5 in both 3′ (N = 3) and 5′ (N = 2) splice sites; at exon 7 in both 3′ (N = 1) and 5′ (N = 2) splice sites; at exon 8 in both 3′ (N = 2) and 5′ (N = 1) splice sites; in the 5′ splice site of exons 3 and 4 (N = 1 each); and in the 3′ splice site of exons 6 and 9 (N = 2 and 1, respectively). Most of these mutations originate truncated isoforms either by introducing a premature stop codon or by altering the reading frame. As a result, the level of canonical, functional p53 protein is reduced to less than 20% in some cases, with the subsequent downregulation of both *TP53* target genes and signatures of *TP53* pathway activity [96].

The phosphatase and tensin homolog suppressor (*PTEN*) gene provides a second example of aberrant, cancer-driving, splicing isoforms originated from mutations in the splice sites. The gene comprises nine exons and apart from the full-length, canonical isoform, eight AS-derived isoforms were reported early [97]. Intron retention leading to premature transcription termination is the most frequent event, followed by exon skipping, which also causes truncated isoforms and, more recently, two additional translational isoforms arising from upstream initiation codons were described [98,99]. Although the inactivating mutations of the gene have been extensively studied [100], data on mutation at splice sites, which result in oncogenic isoforms, are scarce. Recently, however, Wang et al. [101] reported a study including 172 breast cancer patients who received neoadjuvant chemotherapy. Most of them (N = 104) were chemosensitive, while the remaining 68 patients were chemoresistant. A screening for *PTEN* polymorphisms revealed a significant association between a single-nucleotide polymorphism (SNP) at the 3′ splice site of exon 5 and chemosensitivity of breast cancer. The SNP consists in the A > G mutation in the AG splice site and leads to the use of an alternative 3′ site located at intron 4. This results in the in-frame inclusion of 18 nucleotides as well as in a change in the binding capacity of an ESE. The wild-type allele was associated with chemoresistance, while chemosensitivity was connected with the mutant allele. The molecular cause for chemosensitivity is linked to the increased phosphatase activity of the mutant protein [101]. PTEN is the main negative regulator of the Phosphatidylinositol-4,5-Bisphosphate 3-Kinase Catalytic Subunit Alpha-AKT Serine/Threonine Kinase 1 (PI3K-AKT) pathway because through its phosphatase activity dephosphorylates the 3-phosphoinositide products of PI3K [102]. Based on the known structure of wild-type PTEN, the results of protein docking showed that in mutant PTEN, the interaction with Prolactin-Induced Protein (PIP3) may be stronger, making the dephosphorylation of PIP3 to PIP2 easier, with the final outcome of increased chemosensitivity [101].

We have to remind ourselves that mutations in splice sites occur in introns and, therefore, are not detected when looking at mutations in coding sequences. This underscores the importance of incorporating non-coding sequences into cancer panels for clinical practice or analysis through whole-genome sequencing. Despite potential depth limitations, this approach would enable the identification of variants in these critical locations.

While, in the two previous examples, mutations in cis elements referred to splice sites, there are also cases in which aberrant splicing results from mutations in cis regulatory sequences. The Cyclin-Dependent Kinase Inhibitor 2A (*CDKN2A*) locus displays a singular feature reported almost 30 years ago, when Sidransky showed that it encodes two different transcripts regulated by different promoters [103]. One of them, *INK4A*, consists of the 1α, 2, and 3 exons, while exon 1β is the first exon of *ARF*, which uses an alternative polyadenylation site in exon 2. The reading frames of *INK4A* and *ARF* are different. Accordingly, the amino acid sequences of the encoded proteins p14^ARF^ and p16^INK4A^ are unrelated. Both proteins have tumour suppressor properties, albeit their action follows different mechanisms. The p16/p14 ratio, which is related to metastasis, is affected by mutations at cis regulatory elements, and the following ones have been described in metastatic renal cell carcinoma: G > C at position 21970916, which destroys an ESE Site; C > A 21974781, which destroys an ESE site and generates a new ESS site; and G > T at position 21994232, which also abolishes an ESE site [101]. These elements are located within exons 1α and 1β and thus their mutations influence the nature of the transcript.

In a similar way, mutation G > A in the first exon of Enhancer Of Zeste 2 Polycomb Repressive Complex 2 Subunit (*EZH2*) breaks an ESE site and generates a new ESS site [104], resulting in an aberrant isoform. *EZH2* is a proto-oncogene, comprising 20 exons, which codes for a histone methyltransferase. There are also other factors controlling the AS of the gene which will be dealt with below.

### 3.2. Involvement of Non-Mutated Cis Elements in Alternative Splicing

Splicing-regulatory cis elements may also be involved in AS through the binding of some proteins that compete against SFs. A paradigmatic example has been described for the Tumour Protein P63 (*TP63*) gene, a member of the p53 family, which encodes the conserved transcription factor p63. It contains sixteen exons, and two groups of protein variants, namely TAp63 and ΔNp63, have been described. Isoforms from the former group are transcribed from the canonical promoter in exon 1, while ΔNp63 isoforms use an alternative promoter in exon 4, skipped in TAp63 isoforms, and, consequently, lack the N-terminal transactivation domain (Figure 5). Two possible polyadenylation sites exist: one in exon 12 and the other in exon 16. Within both groups of variants, four possible isoform types exist: α, β, δ and γ. In the first three isoforms, exon 12 is skipped, and, therefore, transcription ends at exon 16. On the contrary, transcription of γ isoforms ends at exon 12 and lack the domains present in the last four exons (Figure 5).

Exons 5–11 are constitutively spliced and, therefore, are present in all the variant types. Exons 13 and 16 are present in all the TAp63 isoforms. The remaining exons are facultative; we have already mentioned that exons 1–3 are skipped in ΔNp63 isoforms, while exons 4 and 12 are skipped in TAp63 variants. Inclusion or skipping of exons 14 and 15 mark the difference between the α, β, and δ isoforms [105]. It has recently been found that the formation of γ isoforms, which are associated with decreased survival at least in patients with head and neck squamous-cell carcinoma [106], is repressed by the polypyrimidine tract-binding protein PTBP1. The polypyrimidine tracts up- and downstream exon 12 are close enough to the 3′ and 5′ splice sites of this exon and the binding of PTBP1, whose level is increased in cancer, blocks the access of U2AF or U1 (Figure 5). The definition of exon 12 is, therefore, hindered, with the consequence of increasing the appearance of β isoforms at the expense of the γ variants [106]. In summary, the binding of PTBP1 to the polypyrimidine tracts competes with the assembly of the splicing machinery to the splice sites of exon 12, finally leading to a diminution in the production of the oncogenic γ variants of TP63.

A somewhat similar mechanism was described for the Estrogen Receptor 1 (*ESR1*) gene, which encodes ERα, the oestrogen receptor α. The primary transcript of the *ESR1* gene starts by 5 alternative exons (1B to 1F). These precede exon 1A, which contains the initiation codon for the full-length protein, usually referred to as ERα66, that is a hallmark of breast cancer and other tumours. The variant ERα46, which antagonises the proliferative activity of the canonical isoform, lacks the N-terminal region of the latter. It is encoded by an *ESR1* transcript that results from the direct splicing of either exon 1E or 1F to exon 2 with the skipping of exon 1A and, therefore, uses a downstream initiation codon, resulting in an N-terminal truncated isoform [107]. It has been found that the high-mobility group A protein 1a (HMGA1a) induces the skipping of exon 1A and, hence, causes the overexpression of ERα46 in MCF-7 breast cancer cells. Ohe et al. have proposed that HMGA1a binds an RNA sequence located 33 nucleotides upstream the 5′ splice site of exon 1A and adjacent to an upstream pseudo-5′ splice site. This binding would preclude the union of the splicing factor U1 with the result of exon 1A skipping [108].

A different mechanism, even though it involves wild-type splice sites, has been described in the *ARID1A* gene. We have previously mentioned the role of ARID1A in the AS of *SAT1* pre-mRNA; here, the AS of *ARID1A* itself is considered. The gene consists of 20 exons, and two transcript variants encoding different isoforms, ARID1A-L and ARID1A-S, have been described. The first one, ARID1A-L, includes the whole exon 18, while the short one results from the usage of an alternative 3′ splice site within the same exon. The shift between both isoforms is controlled by the EWS-FLI1 fusion protein [109]. The mechanism involved in this regulation is a complex one and was elucidated in 2019. ARID1A protein is a member of the BAF remodelling complex and can interact with EWS-FLI1 through a region encoded by exon 18, and so ARID1A-L is the only isoform capable of establishing that interaction. The same authors, taking into account that EWS-FLI1 interacts with multiple splicing factors [109] suggested that the fusion protein is tethered to a U1-binding site in *ARID1A* exon 18 through a splice site regulator [110]. This may explain the role of EWS-FLI1 in switching the AS of *ARID1A* towards the long isoform (Figure 6A) and how, in view of the remodelling activity of BAF, the complex formed by EWS-FLI1, ARID1A-L, and the rest of BAF components activates the transcription of Ewing sarcoma genes in a chromatin-dependent manner. Finally, ARID1A-L is also able to stabilise EWS-FLI1 [110]. In this manner, a feed-forward oncogenic cycle may be established in Ewing sarcoma: EWS-FLI1 induces the formation of ARID1A-L, which, in turn, stabilises EWS-FLI1 and supports oncogenesis (Figure 6B).

### 3.3. Alternative Splicing Caused by the Disruption of Splicing Factor Homeostasis

As shown in Figure 4B, changes in the components of the splicing machinery are the second most frequent molecular cause of altered AS and affect 34 cancer driver genes. These changes may be due to mutations in critical regions of splicing factors, alterations in their level, or even to post-translational modifications. Mutations in components of the splicing machinery as a cause of aberrant AS have been found in four genes among those listed in Appendix A, namely *EZH2*, GNAS Complex Locus (*GNAS*), Patched 1 (*PTCH1*), and Cyclin D2 (*CCND2*). In the two former cases, mutations in both U2AF1 and SRSF2 result in aberrant AS [111,112,113]. The splicing of *EZH2* pre-mRNA exhibits multiple other features that will be dealt with below. Mutation in U1 snRNP causes the production of an alternative, prematurely terminated CCND2 isoform, which activates the oncogenic potential of the gene while inactivating the suppressor gene *PTCH1* as a result of the inclusion of a cryptic exon causing a frameshift and an alternative initiation of translation [114].

Changes in the level of splicing factors are a cause of AS alteration in driver genes even more frequently than mutations. Actually, they affect 29 out of the above-mentioned 34 genes (Figure 4B). These 29 genes are listed in Table 2, in which the responsible splicing factors and the consequences of their change are also included. It can be observed that, in some instances, AS leads to non-oncogenic isoforms, while in most cases, the result of aberrant AS is the production of oncogenic variants. The homeostasis of splicing factors may be disrupted by several causes, including several well-known factors. Oxidative stress, for instance, is known to alter the balance of splicing factors [115,116], and hypoxia displays a marked influence on their level, as will be commented below.

Some of the genes in Table 2 deserve special attention due either to the mechanisms in which the splicing factors are involved or to their physiopathological consequences. This is the case, for instance, of the Von Hippel–Lindau (*VHL*) tumour suppressor gene. VHLα is a translational variant which results from the use of an alternative start site upstream and in frame with the normal ATG codon. The heterogeneous nuclear ribonucleoprotein A2B1 (hnRNPA2B1) interacts with VHLα, but not with other isoforms of VHL [117].

The hnRNPA2 proteins bind to sequences flanking exon 9 of the pyruvate kinase (*PKM*) pre-mRNA, resulting in the inclusion of the alternative exon 10 with a shift to the PKM2 isoform [118]. Consequently, VHLα modulates PKM splicing and inhibits the Warburg effect [117]. The situation is more complex, because the selection of the alternative translational start site that yields VHLα is regulated by hnRNPA2B1. Therefore, the mutual regulatory behaviour involving VHLα and hnRNPA2B1 constitutes an anti-malignant feedback loop [119].

**Table 2 cancers-16-02123-t002:** Driver genes whose AS is affected by the level of splicing factors ^1^.

Gene	Factor	Effects of Upregulating Factor	Physiopathological Effects	Cancer Type	Refs.
*KMT2C*	SRSF3	e45 sk. alt 3′ SS in e46	changes in H3K4me3	A2780 sublines	[120]
*CTNNB1*	hnRNPH1	isoform shift	none	RMS	[121]
*EGFR*	SRSF2	e17 and e18 sk; alt polyA	sensitivity to gefitinib	LUAD	[122]
*CHD1*	ESRP1, ESRP2	normal e14 splicing	no induction of EMT	PCa	[123]
*VHL*	hnRNPA2B1	alt translation of VHLα	inhibition of Warburg effect	ccRCC	[119]
*EP300*	SRSF3	e14 inclusion	sk. e14 promotes tumour	OS	[124]
*AR*	RBM39	cryptic exon	oncogenic isoform AR-V7	PCa	[50]
*MYD88*	SF3A/B	normal e2 splicing	normal TLR signalling	lymphoma	[125,126,127]
*TCF7L2*	TRA2B *	e4 inclusion	oncogenic isoforms	HCC, organoids	[128,129]
*FGFR2*	ESRP1, ESRP2	normal e9 splicing	inhibition of EMT	OSCC	[70]
*EZH2*	SF3B3	e14 inclusion	increased proliferation	ccRCC	[130]
*MAX*	hnRNPA1	inclusion e5, ΔMAX	promotes proliferation	glioma	[131]
*RAC1*	hnRNPA1	no cryptic e3b	RAC1b tumourigenic variant	HeLa	[132]
	SRSF1, RNPS1				[133]
*BCLAF1*	SRSF10	cryptic e5a	tumourigenic variants	CRC	[134]
*FANCD2*	U2 snRNP	proximal polyA	non-malignant	HCT116, RKO	[135]
*PRKCD*	SRSF2	alt 5′ SS in e10	malignant isoform PKCδVIII	SKOV3	[136]
*MAP2K7*	MBNL1	normal e2 splicing	malignant isoform	solid tumours	[137]
*IRF1*	SFPQ	increases e7 sk	decreases IFNγ	Th1 cells	[138]
*FAS*	hnRNPA1, SRSF6, SRSF4	inclusion e6	proapoptotic isoform	HCT116, HeLa	[139,140,141]
*DHX9*	hnRNPM, SRSF3	no poison exon	lower expression of DHX9	Ewing sarcoma	[142]
*SMC1A*	SRSF2	normal splicing	metastasis suppression	CRC	[143]
*KLF4*	SF3B4	increases e3 sk	inactivation of p27^Kip1^	HCC	[144]
*AKT3*	SRSF2	increases e8 sk	over-expressed isoform	H358, HeLa	[122]
*ARAF*	hnRNPH, HNRNPA2B1	full length isoform	cancer progression	GHD-1, HCT116, NIH3T3	[145,146]
*RAP1GDS1*	several putative SFs	e5 sk; SmgGDS-558	increases proliferation	PC3, VCaP	[147]
*MDM4*	RNPS1	e6 splicing; no MDM4-S;	anti-apoptotic	CC	[133]
	hnRNPH1	no IR	none	RMS	[121]
*NUMA1*	MBNL1	normal e16 splicing	no proliferation	BRCA	[53]
*EWSR1*	hnRNPH1	e8 sk	reduction fusion oncogenes	Ewing sarcoma	[148]

^1^ The genes are listed, in decreasing order, according to the number of samples of the cohorts used by Martínez-Jiménez et al. [31] in which mutations occur. * The involvement of TRAB2B has been substantiated only in diabetes. The name of the relevant isoforms is included when required. Abbreviations used in the table: e[No.], exon No.; sk, skipping; SF, splicing factor; alt, alternative; IR, intron retention; SS, splicing site; TLR, Toll-like receptor; OC, ovarian cancer; RMS, rhabdomyosarcoma; HCC, hepatocellular carcinoma; OSCC, oral squamous-cell carcinoma; PCa, prostate cancer; ccRCC, clear-cell renal cell carcinoma; CRC, colorectal carcinoma; LUAD, lung adenocarcinoma; PC, pancreatic cancer; CC, cervical cancer; BRCA, breast cancer. When the research was conducted with cell lines, their common name is given.

The splicing of two genes from Table 2, namely the Rap1 GTPase-GDP dissociation stimulator 1 (*RAP1GDS1*) and the structural maintenance of chromosomes 1A (*SMC1A*), is affected by hypoxia. 152]SmgGDS, the protein encoded by the oncogene *RAP1GDS1*, promotes the activity of RhoA and RhoC GTPases; this is associated with poor survival in cancer patients [149]. *RAP1GDS1* contains 15 exons and can give rise to two splice variants, namely SmgGDS-558 and SmgGDS-607, which differ in the presence of exon 5, skipped in the former isoform [150,151]. Both variants are oncogenic, but the SmgGDS-607/SmgGDS-558 ratio is higher in breast and lung cancer, so that targeting the switching between isoforms has been proposed as a therapeutic approach in these cancers [152]. [Bowler et al. found that hypoxia favours the skipping of exon 5 and shifts the isoform balance towards smgGDS-558 in prostate cancer cells [142]. Hypoxia triggers different responses, one of them being the stabilisation and import of the inducible factor α (HIF1α) into the nucleus. This factor induces the upregulation of miR-222-3p that, in turn, inhibits SRSF2 expression at a post-transcriptional level by binding its 3′ UTR, finally resulting in the aberrant expression of Vascular Endothelial Growth Factor A (*VEGFA*) isoforms in breast cancer [153]. A different mechanism has recently been described for *SMC1A*. In this case, a non-coding small RNA tRNA-derived fragment (tRF) regulates the expression of the lncRNA *MALAT-1* (metastasis-associated lung adenocarcinoma transcript 1), which, in turn, interacts with SRSF2. The final issue is the abnormal expression of several *SMC1A* isoforms, associated with an increased propensity for metastasis in CRC [143].

A different serine/arginine-rich splicing factor, namely SRSF6, is also affected by hypoxia. Its level is reduced under severe hypoxic conditions by the SRSF4-induced inclusion of a poison exon. This diminution in the level of SRSF6 causes a reduction in splicing activity, because under normoxic conditions, SRSF6 binds either to the alternative exons or upstream of 3′ splice sites, thus favouring the inclusion of these exons [154]. This SRSF6 depletion in hypoxia may affect the splicing of many genes and explain the adaptation of cancer cells to hypoxic conditions.

Post-translational modification of splicing factors is another change that may potentially influence the splicing of driver genes. We have found that acetylation of hnRNPA1 and L was induced in response to Epidermal Growth Factor (*EGF*) in some *KRAS* mutants [155] and, under these conditions, the splicing of several genes is altered [151]. The aberrant AS two of these genes, namely Zinc Finger Protein 518B (*ZNF518B*) and Ependymin Related 1 (*EPDR1*) correlates with their oncogenic characteristics [156,157,158,159], but, to date, no evidence exists for the alteration of the AS of cancer driver genes driven by the post-translational modification of splicing factors.

### 3.4. Epigenetic and Chromatin-Associated Causes of AS

The idea that some epigenetic marks may be considered cancer hallmarks is a long-standing one [160], and the relationships between epigenetics and aberrant splicing in cancer also constitute a well-defined subject [33]. Here, we revise the influence of epigenetics on splicing of cancer driver genes and, taking into account that epigenetic changes occur in a chromatin scenario, the structural changes in chromatin that determine splicing will also be considered. The most common epigenetic factors involved in the mechanism of AS in driver genes are DNA methylation and histone modifications.

DNA methylation is a particularly interesting epigenetic modification due to two facts. First, it was shown some time ago that alterations in DNA methylation may occur prior to the appearance of mutations in driver genes. For instance, the hypermethylation of the O-6-methylguanine-DNA methyltransferase (*MGMT*) gene, which encodes a DNA repair protein, precedes *KRAS* mutations in the onset of colorectal cancer [161,162], even when mutations in *KRAS* are early events in carcinogenesis. More recently, a genome-wide study revealed that in the transit from adenomatous hyperplasia to invasive lung adenocarcinoma, alterations in DNA methylation precede mutational events [163]. Secondly, DNA methylation is a targetable modification [164], and this offers a promising therapeutic possibility for early-detected tumours.

#### 3.4.1. Epigenetic Causes of Selection of Alternative Promoters

DNA methylation and histone modifications are the main factors governing the selection of alternative promoters. The hypermethylation of promoters inhibits transcription initiation and several histone marks also result in the acquisition of a chromatin structure, hindering transcription. On the contrary, other histone modifications are associated with a chromatin structure competent for transcription (Figure 7A).

The *CDKN2A* locus has been mentioned previously, but in the present context, it should be mentioned that, at the end of the last century, it was already known that the promoter of *INK4A* is hypermethylated in several cancers [165], favouring the alternative transcription of *ARF*. This was the first example of how an epigenetic modification may alter the ratio of different transcripts from the same gene.

Another example of the regulation of isoform level by intragenic DNA methylation is given by the positive regulatory domain 1 (*PRDM1*) gene. Two isoforms, namely PRDM1α—the full-length isoform—and PRDM1β, are encoded by the gene. The latter isoform is transcribed from an alternative promoter located in intron 3 and part of exon 4, and the PRDM1β protein lacks 101 amino acids from the N-terminus, resulting in the loss of part of the positive regulatory domain that catalyses the methylation of H3K9. As an intact domain is required for the tumour suppressor activity of PRDM1α, PRDM1β displays oncogenic properties and is overexpressed in several haematological malignancies (for a review, see [166]). Zhang et al. [167] analysed the methylation of CpG islands located in promoters of the α and β isoforms in diffuse large B cell lymphoma patients and B lymphoma cell lines and concluded that aberrant methylation-driven silencing of PRDM1α and activation of PRDM1β by hypomethylation of its alternative promoter are frequent events in diffuse large B cell lymphoma.

The AKT Serine/Threonine Kinase 3 (*AKT3*) gene encodes a serine/threonine protein kinase involved in tumorigenesis. A novel variant of this gene, originated from an alternative transcriptional start site unique to tumours, was described in human papillomavirus-positive oropharyngeal cancer patients. Guo et al. analysed the methylation status of DNA surrounding this novel start site in a patient cohort including 46 tumours and 25 normal samples. They found that, while this region is highly methylated in normal samples, its level of methylation is very low in tumours and even null in patients showing alternative splicing events in the *AKT3* gene [168]. These results also show how intragenic DNA methylation may be a decisive factor in the selection of an alternative oncogenic isoform.

Alternative usage of two promoters and AS of the forkhead box P1 (*FOXP1*) gene are also involved in haematological malignancies. Human *FOXP1* contains twenty-one exons and at least two alternative promoters may be used. The first five exons are non-coding, and the promoter used by the canonical, full-length isoform FOXP1L originates transcripts starting in exon 6 or in the alternative exon 6b, while the use of an alternative promoter results in shorter transcripts, starting in exon 8. The skipping of exons 8, 9, or 10 gives rise to additional isoforms [169]. Treatment of diffuse large B cell lymphoma cell lines with 5-azacytidine alters the expression of short oncogenic isoforms, suggesting that methylation of the internal promoter regulates its activity [170].

A last example of DNA methylation-directed usage of alternative promoters is provided by the positive regulatory domain 2 (*PRDM2*) gene. Four different transcripts and two protein isoforms have been described, namely PRDM2a and PRDM2b, which are often named RIZ1 and RIZ2 after the alias of the gene (Retinoblastoma-Interacting Zinc finger). RIZ1 is the full-length variant, with 1718 amino acids; RIZ2, about 200 amino acids shorter, results from the translation of transcript 3, which uses an internal promoter (promoter 2) and lacks the N-terminal sequences of RIZ1, in which the PR/SET domain, containing the histone methyltransferase activity, is located. This circumstance is the cause of the discrepant function of both protein isoforms. While RIZ1 inhibits cell growth, RIZ2 promotes proliferation and is the main isoform found in cancer tissues [171]. It was known that CpG methylation in *PRDM2a* promoter inhibits the expression of *RIZ1* (for a review, see [172]) and that this DNA modification switch the RIZ1/RIZ2 ratio towards the oncogenic variant. Rienzo et al. have recently hypothesised that the inhibition of CTCF binding to its methylated CpG-containing target sites may also cooperate with the regulation of *RIZ1* expression [173]. At any rate, it is clear that DNA methylation is a decisive factor in the selection of the predominating PRDM2 isoform.

A somewhat different consequence of DNA methylation in the selection of variants is provided by the core-binding factor, runt domain, alpha subunit 2; translocated to, 3 (*CBFA2T3*) gene. This gene participates in translocations, involving immunoglobulin loci, which are associated with lymphoid malignancies. Among these, Burkitt lymphoma, the most frequent B cell lymphoma in children, is especially aggressive and has been the subject of many studies. This lymphoid neoplasm is thought to derive from germinal-centre B cells. The AS events of *CBFA2T3* are different in Burkitt lymphoma as compared to germinal centre B cells. Of note, isoform 2, which lacks two regions of the canonical form, is preferentially expressed in Burkitt lymphoma. This variant regulates cAMP-mediated signalling. Methylation around the promoter of *CBFA2T3* isoform 1, an epigenetic mark characteristic of Burkitt lymphoma, negatively correlates with the expression of the canonical isoform. Furthermore, the IG translocation breakpoints in *CBFA2T3* disrupt isoform 1 and both factors favour the expression of isoform 2 [174].

The ten-eleven translocation methylcytosine dioxygenase 1 (*TET1*) gene encodes an enzyme that catalyses the first steps of DNA demethylation, initiated by the oxidation of 5-methylcytosine to 5-hydroximethylcytosine. The gene contains twelve exons, and two protein isoforms have been described. The transcription of the canonical, full-length isoform starts at exon 1, although the translation start codon is located at exon 2, which contains the CXXC domain. This domain binds to unmethylated stretches of CpG islands, a function that has been a controversial issue. The canonical isoform, TET1^FL^, was first characterised and found to be expressed especially in embryos. In 2017, Good et al. [175] described the second, novel isoform, TET1^ALT^, which is aberrantly expressed in multiple cancer types such as breast, uterine, and glioblastoma. This isoform is transcribed from an alternate promoter in intron 2 and its start codon is located in exon 4. Therefore, TET1^ALT^ lacks the CXXC domain, and this circumstance allowed the authors to postulate a role for this controversial domain in the protection of unmethylated CpG islands during embryo development. The failure of TET1^ALT^ to play this role may account for its oncogenicity. For the purposes of the present review, an interesting finding of Good et al. is that in GM12878, a lymphoblastoid cell line, the TET1^ALT^ promoter is enriched in H3K4me3, a mark characteristic of active chromatin, while the TET1^FL^ promoter is not. Contrariwise, in embryo stem cells, H3K4me3 is abundant in the canonical promoter but not in the alternate one. An opposite result was obtained for the repressive mark H3K27me3, which is abundant in the TET1^FL^ promoter in GM12878 cells, but not in that of TET1^ALT^ [175]. These results show that the selection of *TET1* isoforms is directed by histone epigenetic marks.

Histone methylation also controls the AS of anaplastic lymphoma kinase (*ALK*) and of fibroblast growth factor receptor 2 (*FGFR2*) genes. The mechanism involved in the first case is similar to that of *TET*, namely, apart from the canonical promoter that controls the transcription of the full-length isoform, an alternate promoter exists at intron 19. The usage of this promoter results in the expression of the isoform ALK^ATI^. Three in-frame start codons exist in the *ALK^ATI^* transcript, resulting in three protein isoforms from which the N-terminal region of the canonical isoform, which contains the extracellular and transmembrane domains, is missing. The kinase catalytic domain is present in the ALK^ATI^ proteins; consequently, they promote uncontrolled, growth-factor-independent phosphorylations, resulting in cell proliferation and tumorigenesis. The *ALK^ATI^* promoter is regulated by histone methylation: in *ALK^ATI^*-expressing tumours, the alternate promoter is enriched for H3K4me3 [176]. As mentioned above, *TP63* may be transcribed to ten different isoforms; five of them, which are truncated at the 5′ terminus, collectively receive the name of ΔNp63 and are especially expressed in several tumours. The *TP63* chromatin is enriched in H3K27ac, a mark favouring transcription, in the vicinity of the transcription start sites of the full-length and truncated isoforms in a cell type-dependent manner; this may be one of the causes of the selection of the alternate promoter for truncated isoforms [105].

#### 3.4.2. Influence of Epigenetic Modifications on the Assembly of the Splicing Machinery

Apart from their role as regulators of chromatin organisation, epigenetic modifications of histones may influence exon definition, i.e., the selection of the ends of exons that have to be spliced (Figure 7B).

As shown in Table 2, the RNA-Binding Motif Protein 39 (RBM39) splicing factor stimulates the inclusion of a cryptic exon containing a stop signal in the androgen receptor (*AR*) gene, giving rise to the oncogenic variant AR-V7, the best-characterised AR variant, in prostate cancer [50]. The AR-V7 truncated protein is constitutively active in transcription, independently of androgens binding. In this way, AR-V7-positive patients of metastatic castration-resistant prostate cancer are refractory to androgen deprivation therapy (see Brown et al. [177] for a review). In consequence, many efforts have been devoted to elucidating the mechanisms involved in AS events resulting in the production of AR-V7. Apart from the above-mentioned role of RBM39, the involvement of epigenetic factors is of special relevance. It was known since 2014 that under androgen deprivation therapy, several RNA splicing factors are recruited to the alternative 3′ splice site next to a cryptic exon, instead of the 3′ site of the canonical exon [178], but the specific mechanism remained unknown until the work of Duan et al. [179]. These authors found that Lysine Demethylase 4B (KDM4B) plays a crucial role in the selection of the AR-V7 isoform among the 22 variants of the AR gene. RNA-immunoprecipitation assays showed that KDM4B strongly binds the intron near the 5′ end of the cryptic exon and that, in response to androgen deprivation, the histone demethylase associates with general splicing factors, such as the Splicing Factor 3b Subunit 3 (SF3B3) and hnRNPA1. Specifically, the binding of SF3B3 to KDM4B is regulated by the phosphorylation of the latter protein. Co-immunoprecipitation experiments also revealed that Tripartite Motif Containing 28 (TRIM28) interacts with KDM4B and, taking into account that TRIM28 is known to be associated, via heterochromatin protein 1 (HP1), with chromatin carrying the repressive mark H3K9me3, these authors have proposed a model in which the phosphorylation of KDM4B under androgen deprivation conditions triggers the binding of SF3B3, a component of the spliceosome. As KDM4B binds RNA 5′ to the cryptic exon, the spliceosome is tethered close to this exon. On the other hand, the interaction of KDM4B with TRIM28 allows the demethylase to remove the repressive mark H3K9me3 and to open the chromatin in the vicinity of the cryptic exon, which may then be included [179].

In the case of the fibroblast growth factor receptor 2 (*FGFR2*) gene, the situation is more complex. The gene contains two mutually exclusive exons between canonical exons 7 and 9, which were termed IIIb and IIIc by Sanidas et al. [180]. These authors proposed a model in which the RNA processing regulator IWS1 and the kinases AKT1 and AKT3 play a crucial role in the selection of exons IIIb or IIIc, mediated by the introduction of the epigenetic mark H3K36me3 in the chromatin of *FGFR2* gene. According to this model, when IWS1, which is bound to the C-terminal domain of RNA polymerase II, is phosphorylated by AKT3 and AKT1 recruits the methyltransferase SETD2 to the polymerase complex. The recruitment of SETD2 results in the trimethylation of H3K36 during transcription. The nucleosomes of *FGFR2* are then bound to the chromodomain-containing protein MRG15, which recognises H3K36me3 and recruits the polypyrimidine tract-binding protein PTB. In this way, H3K36me3 performs its role as a splicing regulator [84] and exon IIIc is included [180].

#### 3.4.3. Histone Modifications in the Selection of Mutually Exclusive Exons

The last example given in this section is that of the *KRAS* gene, which belongs to the RAS family of small GTPases involved in cell proliferation. It is widely known that *KRAS* encodes two splice variants, KRAS4A and KRAS4B, which result from the inclusion of one of the two mutually exclusive exons, 4A and 4B. Mutations in either exon 2 or 3 result in the constitutive activation of the products and, hence, in increased cell proliferation, so that mutations in *KRAS* occur in many tumours. Both isoforms are activated by mutations, but some differences have been found between the differential functions of the *KRAS* splice variants in the onset and progression of cancer (for a recent review, see [181]). The mechanisms deciding the KRAS4A/KRAS4B ratio were largely unknown, but we found that epigenetic modifications of histones are, at least in part, responsible for the splicing differences. Actually, significant alterations in the level of H3K4me3, H3K27me3, H3K36me3, H3K9ac, H3K27ac, and H4K20me1 at exons 4A and 4B were found between colorectal cancer cell lines preferentially expressing one or other isoform. That some of these differences are responsible for the isoform selection was ascertained by the fact that inhibition of histone deacetylases or of the EZH histone methyltransferase results in alteration of the KRAS4A/KRAS4B ratio [182].

### 3.5. Clinical Relevance of the Aberrant Alternative Splicing in Driver Genes

The existence of several small-molecule inhibitors targeting the splicing factors illustrates the clinical importance of alternative splicing in driver genes for drug development. For instance, pladienolide analogues exhibit specific activity against various SF3b subunits and spliceosomal associated proteins [183]. One notable drug is FR901464 (FR), which specifically inhibits SF3B1. Studies in colorectal cancer have shown that FR treatment affects critical genes such as BRCA, indicating that combining FR with other anticancer drugs, like the PARP1 inhibitor olaparib, can increase sensitivity in BRCA 1/2 deficient cells and produce synergistic effects [184]. Additionally, the natural product FR901464 and its methylated derivative spliceostatin A inhibit splicing in vitro, causing pre-mRNA to accumulate by binding to SF3b. This interaction impacts proteins like p21 and p27, which inhibit the transition from the G1 to S phase by binding to the CDK2 complex; a truncated form of p27 also accumulates in cells treated with FR [185]. Another example is SRPIN340, which targets alternative splicing by regulating MKNK2 through SRPK1 and SRPK2, directly phosphorylating SRSF1 and promoting its transport into the nucleus [186]. These examples highlight the significant potential for developing targeted therapies that exploit alternative splicing mechanisms to improve cancer treatment outcomes. Actually, the pharmacological correction of splicing errors is a promising possibility and some compounds are in advanced preclinical stages [187,188,189,190].

## 4. Conclusions and Future Prospects

Classically, the activation of a proto-oncogene or the inactivation of a suppressor is attributed to mutations in the gene body, but the data given in this review show that, in many instances, aberrant AS may play a similar role. The causes of aberrant splicing are mainly found in mutations of cis elements involved in splicing (Figure 4B). These elements are often found in non-transcribed introns. In other instances, a mutation in exonic enhancers or silencers may be a synonymous one, but with consequences in the binding of the corresponding splicing factor. These considerations should urge the consideration of these possible mutations as causes of cancer. A systematisation of these mutations and their inclusion in novel cancer panels would be an important stage in the management of cancer.

We have entered the era of precision medicine in cancer, and the involvement of aberrant splicing must be considered both for the diagnosis and stratification of patients. A more profound knowledge of the implications of aberrant splicing in cancer may also expand the present therapeutic resources, which have been outlined above.

A field in which more research is needed are the implications of epigenetic dysregulation in aberrant splicing. The ample relationships between epigenetics and splicing [33] as well as the existence of many epigenetic drugs (see, for instance [164]) make this issue a hopeful challenge.

## Figures and Tables

**Figure 1 cancers-16-02123-f001:**
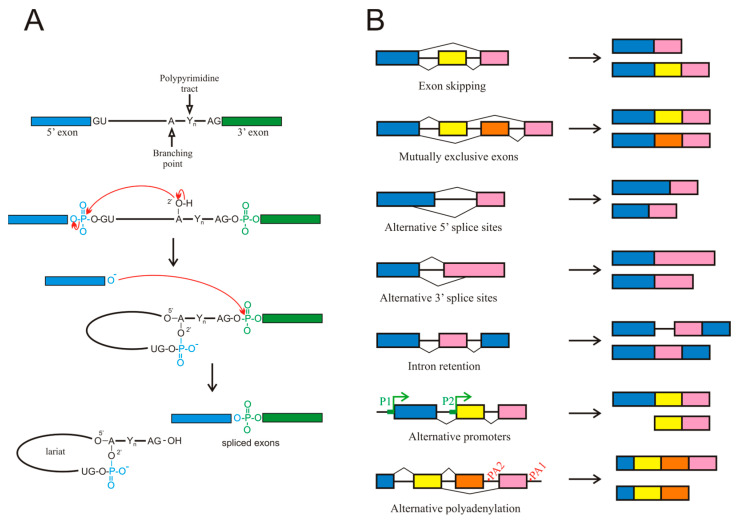
(**A**) Chemical mechanism of splicing. (**B**) Different types of alternative splicing events. Although alternative promoters (P1, P2) or polyadenylation sites (PA1, PA2) are not, strictly speaking, alternative splicing events, they are included because they give rise to different isoforms from a single gene.

**Figure 2 cancers-16-02123-f002:**
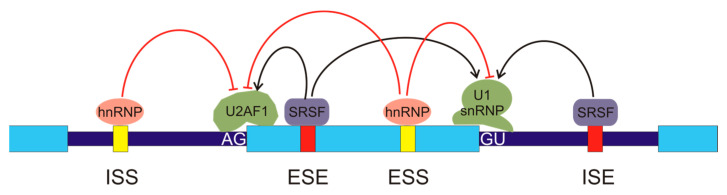
Regulation of alternative splicing. The exons are represented by light blue boxes and the introns are depicted as thick, dark blue lines. There are several cis-regulatory elements in the pre-mRNA: intronic splicing silencers (ISSs), exonic splicing enhancers (ESEs), exonic splicing silencers (ESSs), and intronic splicing enhancers (ISEs). Several types of regulatory trans-elements may bind these sequences. They belong to two main protein families, the heterogeneous nuclear ribonucleoproteins (hnRNPs), which bind the silencers (represented in yellow), and the serine/arginine-rich splicing factors (SRSFs), which bind the enhancers (red boxes). The binding of these factors inhibits or activates, as indicated by the red or black arrows, the recruitment of components of the spliceosome to the AG or GU sequences in the exon borders, thus allowing for exon definition.

**Figure 3 cancers-16-02123-f003:**
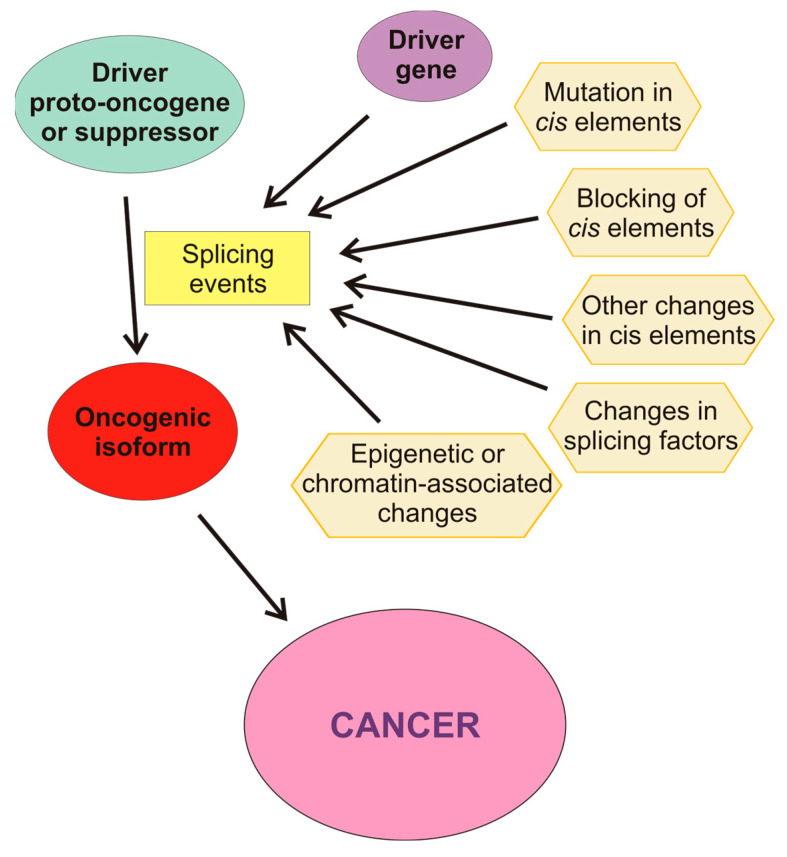
Molecular causes influencing aberrant splicing. The driver proto-oncogene or suppressor may be activated or inactivated by splicing events caused by other drive genes or by causes affecting the cis-regulatory elements of the pre-mRNA: mutations, blocking, or other changes in these elements. Changes in the level of active splicing factors may also cause aberrant splicing. Finally, epigenetic or chromatin-associated alterations of the gene may change the selection of alternate promoters of influence the splicing machinery.

**Figure 4 cancers-16-02123-f004:**
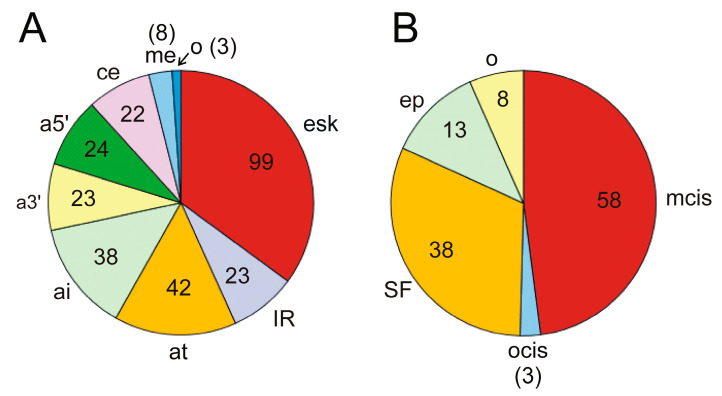
(**A**) Splicing events occurring in the 168 cancer driver genes that are activated by alternative splicing (see Appendix A). esk, exon skipping; IR, intron retention; at, alternative termination; ai, alternative transcriptional or translational initiation; a3′, alternative 3′ splice site; a5′, alternative 5′ splice site; ce, cryptic exon; me, mutually exclusive exons; o, other causes. The pie diagram has been constructed according to the number of genes in which these events occur, i.e., if several events of the same nature are found in a single gene, only one occurrence is considered. (**B**) Distribution of molecular causes of splicing alterations in cancer driver genes. The pie diagram depicts the proportion of these causes in the 97 genes from the Appendix A in which these causes are known and reported in the literature cited. mcis, mutation in cis elements; ocis, other causes affecting cis elements; SF, changes in the level of active splicing factors, including inactivating mutations or dysregulation of their synthesis; ep, epigenetic or chromatin-associated causes; o, other causes. See the text for further details. The number of genes is given within the sectors or in brackets.

**Figure 5 cancers-16-02123-f005:**
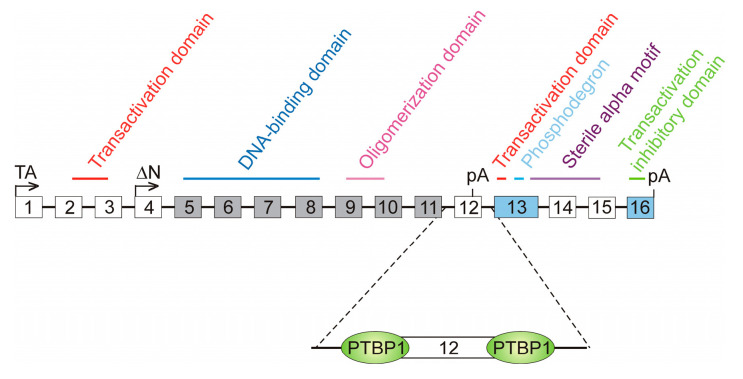
The role of polypyrimidine tracts in the alternative splicing of the *TP63* gene. The 16 exons of the gene are represented as boxes (not drawn to scale) and the different domains present in the resulting proteins are marked; some of them are encoded by two or more exons. Two transcription start sites are possible, one in exon 1, from which TAp63 variants are transcribed, and other in exon 4, which gives rise to ΔNp63 isoforms. Boxes filled in grey represent constitutive exons, present in all the transcript variants, while those filled in blue stand for the exons present in all the TA isoforms. Empty boxes are facultative exons, skipped in some variants. There are two polyadenylation (pA) sites in exons 12 and 16. Exon 12 is skipped in the γ variants of both TAp63 and ΔNp63 groups. Gamma variants lack, in consequence, the C-terminal region of the protein. The differential skipping of exons 14 and 15 gives rise to the α, β, and δ variants in both groups. The regulation of exon 12 skipping depends on the binding of the polypyrimidine tract-binding protein, PTBP1, to its sites at both sides of the exon. Binding of PTBP1 hampers that of U2AF1 and U1 to the splice sites and so leads to the exclusion of exon 12 and the inhibition of carcinogenic γ variants.

**Figure 6 cancers-16-02123-f006:**
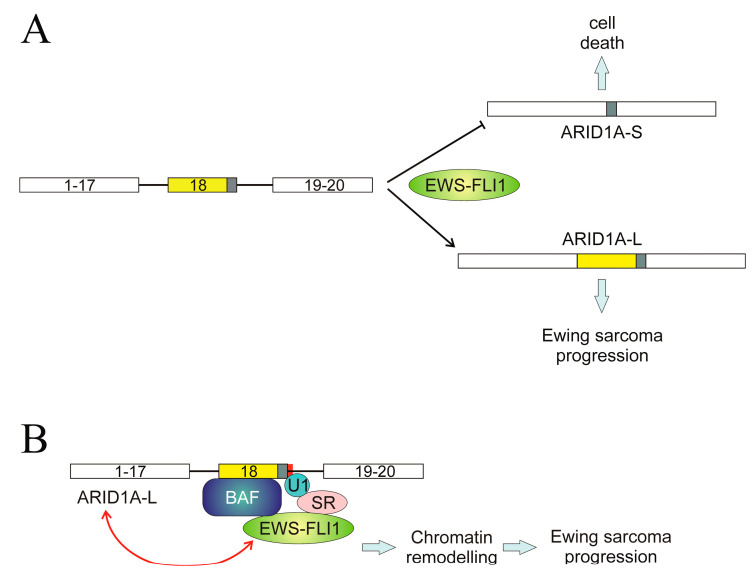
Regulation of *ARID1A* splicing in Ewing sarcoma. (**A**) The fusion protein EWS-FLI1 induces the inclusion of the whole exon 18, resulting in the production of the oncogenic, long isoform ARID1A-L and inhibits the use of an alternative 3′ splice site in exon 18, which would give rise to the apoptotic, short isoform ARID1A-S. (**B**) A possible mechanism for the role of EWS-FLI1. This fusion protein, through its binding to a splicing regulator, is anchored to the spliceosome at the 3′ end of exon 18. As ARID1A-L is a member of the BAF remodelling complex, this results in the chromatin-mediated activation of genes involved in the progression of Ewin sarcoma. EWS-FLI1 causes the formation of ARID1A-L, which, in turn, stabilises the fusion protein, leading to a feed-forward oncogenic cycle. The figure is a re-interpretation of the data obtained by Taylor et al. [107].

**Figure 7 cancers-16-02123-f007:**
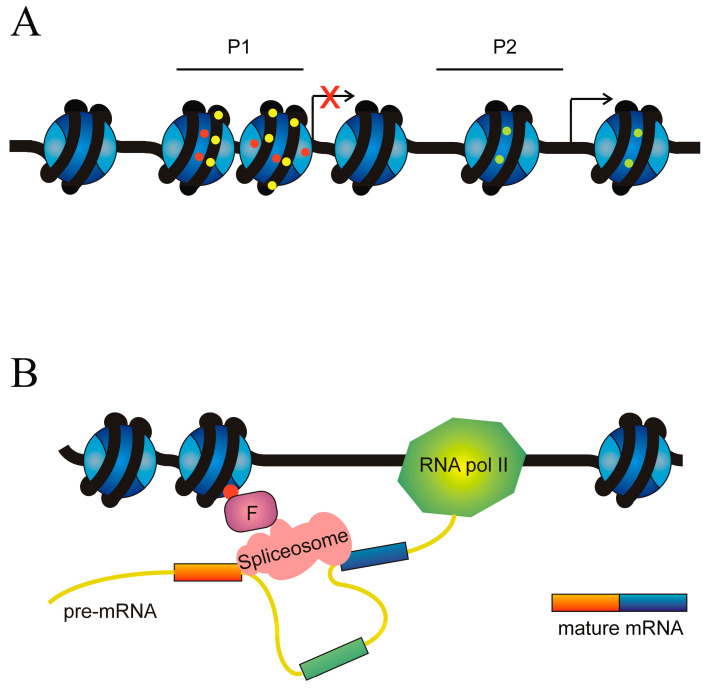
Role of epigenetic factors in the alternative splicing of cancer driver genes. (**A**) Epigenetic causes of the selection of alternative promoters. An ideal gene with two alternative promoters, P1 and P2, is depicted. Transcription from P1 is inhibited by either DNA hypermethylation (yellow circles on DNA) or repressive marks on histones (red circles), which cause the adoption of a heterochromatic chromatin structure. Transcription from P2 is allowed by both the absence of DNA hypermethylation and the presence of histone permissive marks (green circles). (**B**) Epigenetic marks influence the recruitment of the spliceosome. The figure represents an ideal gene in which an epigenetic mark (red circle) on the histones recruits, through a factor F, the spliceosome in a way that results in the skipping of the exon (coloured green). In the absence of F, definition of the exon (coloured green) would result in its inclusion in the mature mRNA.

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
