# Peer review of "The Many Roads from Alternative Splicing to Cancer: Molecular Mechanisms Involving Driver Genes"

_cancers, 2024, doi:10.3390/cancers16112123_

Round 1

Reviewer 1 Report

Comments and Suggestions for Authors

Specific comments to the authors

The submitted review "The Many Roads from Alternative Splicing to Cancer. Molecular Mechanisms Involving Driver Genes" provides a comprehensive overview of alternative splicing in relation to oncogenesis via modulation of cancer driver genes, based on previously published reviews as well as in vitro and in vivo experiments.

Topics covered range from "basic" mechanisms of alternative splicing to the importance of driver genes that gain oncogenic potential through alternative splicing. In summary, the author provides an interesting overview of the possible role of alternative splicing in human carcinogenesis, which is mostly easy to read, follow and understand. The authors should clarify some aspects before accepting the manuscript for publication, as mentioned below.

 # Abstract: The abstract does not contain a real conclusion.

# Table 1: The content should be sorted or divided by 'role'. In addition, the associated cancer entities should also be added by the authors. The "role" of alternative splicing genes could be used as a subheading for Chapter 2.2 "2.2. Driver genes that control alternative splicing of other genes".

# Figure 4: It is not clear where the data in Figure 4b come from. Please clarify.

# Chapter 3.1-3.: Some specific examples of alternative splicing in driver genes: The clinical relevance of this chapter should be more strongly emphasised by the authors. What is the real transfer for the development of specific drugs in relation to alternative splicing?

# Table 2: The authors should sort the table by the variable "gene". The authors should also add the pathophysiological effects.

# Chapter 3.4.: The epigenetic and chromatin-associated causes of AS: The known findings on how DNA methylation and histone modifications influence and/or modulate alternative splicing should be presented in an additional figure.

# Chapter 4.: Conclusions and future perspectives: This chapter is largely speculative and should include some kind of milestones.

Comments on the Quality of English Language

Minor editing of English language required.

Author Response

Reviewer #1

Thank you very much for your useful suggestions, which we have dealt with as follows:

# Abstract: The abstract does not contain a real conclusion. We have modified the last lines to include a conclusion.

# Table 1: The content should be sorted or divided by 'role'. In addition, the associated cancer entities should also be added by the authors. The "role" of alternative splicing genes could be used as a subheading for Chapter 2.2 "2.2. Driver genes that control alternative splicing of other genes". The content of the table has been sorted according to the role of each gene in splicing. Using this role as subheadings in the main text would complicate the organization of the section, in which we wish to emphasize the mechanism of action of the different genes in the overall AS process.

# Figure 4: It is not clear where the data in Figure 4b come from. Please clarify. The data are those reported in the literature cited, as we mention in the revised figure caption.

# Chapter 3.1-3.: Some specific examples of alternative splicing in driver genes: The clinical relevance of this chapter should be more strongly emphasised by the authors. What is the real transfer for the development of specific drugs in relation to alternative splicing? A novel section, “3.5. Clinical relevance of the aberrant alternative splicing in driver genes” has been included. Research on these drugs is still in a preclinical stage.

# Table 2: The authors should sort the table by the variable "gene". The authors should also add the pathophysiological effects. The table foot now explains the rationale of gene sorting. As to the pathophysiological effects, you are right. We have changed the title of the corresponding column of the table, because it describes, in our opinion, the main observable effects from a pathological perspective. For instance, cancer progression effects or pro-apoptotic ones, effects on EMT, or cases like the Warburg effect, which have direct implications for the pathology of the disease. As most of these questions are discussed in the text, we think that including more information in the table would be embarrassing.

# Chapter 3.4.: The epigenetic and chromatin-associated causes of AS: The known findings on how DNA methylation and histone modifications influence and/or modulate alternative splicing should be presented in an additional figure. A new figure has been included in the revised manuscript. We are especially indebted to Reviewer #1, because after preparing this figure we realized that the whole section may be arranged in a different way, namely, according to the mechanisms of action of the different epigenetic modifications instead of the chemical nature of these modifications (DNA methylation and histone post-translational modifications). This prompted us to reorganize the whole section, without adding novel information. We think that this new arrangement fits much better with the purpose of the review.

# Chapter 4.: Conclusions and future perspectives: This chapter is largely speculative and should include some kind of milestones. Two ideas for future research have been included.

Reviewer 2 Report

Comments and Suggestions for Authors

Francisco Gimeno-Valiente et al. in “The Many Roads from Alternative Splicing to Cancer. Molecular Mechanisms Involving Driver Genes” show in extremely depth how aberrant alternative splicing is the key factor in the oncogenic potential of driver genes. There are various possibilities:
- it is possible to act directly on alternative splicing through genes that cause the deregulation of the splicing of other genes, through a cascade effect;
- it is possible to act at the level of aberrant alternative splicing of a proto-oncogene or a tumor suppressor gene, which can determine the onset of an oncogenic variant.
The authors showed how the oncogenic potential of 199 genes is derived from aberrant alternative splicing, furthermore the molecular mechanisms of 40 genes were shown. The authors suggest that a deeper understanding of the implications of aberrant splicing in cancer could improve current therapeutic strategies for numerous malignancies. I consider original the proposal of pharmacological correction of errors caused by alternative splicing as a promising possibility for cancer treatment. The references are appropriate and recent. They support the conceptualizations present in the review. To improve Table 2, the authors should also explain the name of the reference cell lines for the tumor types being studied. The authors should better clarify the causes of alternative splicing, and the role of “in cis or in trans”, line 15.

Comments on the Quality of English Language

Minor editing of English language required

Author Response

Reviewer #2

Thank you very much for your comment on table 2, in which the name of the cell lines has been added. A clarification of the meaning of “in cis or in trans” has also been added.

Reviewer 3 Report

Comments and Suggestions for Authors

Review on the manuscript titled “The Many Roads from Alternative Splicing to Cancer. Molecular Mechanisms Involving Driver Genes” by Gimeno-Valiente et al., 2024

                The authors compiled a review on the issue of splicing impact in cancer etiology. They “collected a list of 568 genes drivers of cancer and revised the literature to select those involved in the alternative splicing of other genes as well as those in which its pre-mRNA is subject to alternative splicing”.

By publications research, they compiled 31 Splicing Factors (SF) genes list subject to regular mutations leading to tumor, and the cancer driver genes which mutations manifested in their aberrant splicing routine leading to a disease (168 genes).

Lastly, the provide “Details on the mechanisms of alternative splicing-driven activation of proto-oncogenes or tumour suppressor genes are given for more than 40 driver genes”.

Based on the schema above, the authors presented the basic features of splicing routine (Fig. 1A, B), as well as SF typical binding pattern (Fig.2) along with transcription-coupled Pol II mediated exons ‘mapping’. Authors mention alternative promoter/polyadenylation sites impact in shaping transcript isoform,

Fig. 3 outlines the schematics of the work featuring subjects the authors address further, including Proto-oncogenic Driver gene and splicing aberrant impacts leading to the  oncogenic isoforms. The authors listed the possible aberrations in splicing pathway factors that may lead to the aberrant splicing of the particular gene of interest.

Chapter 1.3 addresses the cancer driver genes etiology, mentioning two opposite oncogenic genes classes as oncogenes and tumour suppressor genes. They mentioned the recent paper outlining 568 cancer driver genes.

After defining major terms and the classes, the authors proceed to present the review of cancer driver genes vs splicing instances.  The authors present the basic spliceosomal classes mentioned in cancer driver list in Table 1. They specifically indicated that SF3B1 splice factor bears increased mutability thereby often becoming the driver gene. Notably, the chromatin remodeling machinery actively participates in splicing outcome as reported earlier and observed in Table 1. The authors justly paid specific attention to the issue since it’s now become essential consideration in splicing outcome.

The authors then present the Supplementary Table S1 displaying a list 168 cancer driver genes aroused by splicing event, together with the relevant AS events and mechanisms involved in the splicing upon availability. It is supplemented with Fig. 4 with pie charts of splicing type  distribution instances in 168 drivers, and the stat on the causes of malicious events in them (mutation on cis elements/SFs homeostatic misbalance/epigenetic or chromatin-associated cause, and share of non-specified). 50% of mis-splicing attributed either to disruption of SF binding sites in driver genes. Mutations in SF themselves observed in 30% of cases. Lastly, chromatin modifications responsible for 10%.

Chapter 3 is devoted to the specific examples of both SF/driver genes mutations and their elaboration.

Overall, the review is well composed and addresses rather novel knowledge on splicing phase impact, including chromatin remodeling. Some notes are presented below.

1)      P.10/393: “or splicing regulatory elements (ESEs, ISEs, ESSs or ISS” -> or splicing regulatory elements BINDING SITES (ESEs, ISEs, ESSs or ISS)

2)      Title: “3.3. Alternative splicing caused by changes (misbalance?) in splicing factors” – May it be made more specific on “changes”? I suggest: “3.3. Alternative splicing caused by disruption in splicing factors homeostasis/balance/expression level”, just it should be what you mean by that.

3)      Fig. 4 A: It would be worth assessing if the AS types distribution is shifted from the whole genome stats. You may find it in an appropriate paper. It looks like it does not.

4)      The authors rightly state that “Changes in the level of splicing factors is a cause of AS alteration in driver genes EVEN (add) more frequent than mutation”. It’d be good to elaborate on the possible causes of levels alteration. I’d like to state the SFs homeostasis may be disrupted by the environmental stress. For example, ischemic insult shuts neuron expression rate close to negligent level, and the SFs expression rate goes down not proportionate, e.g. homeostasis will be disrupted. So, many types of stresses may lead to malignancy.

5)      English maybe worth some polishing to a reasonable extent.

Comments on the Quality of English Language

Some points/phrases/titles are not sufficiently explicit, more information is advisable.

Author Response

Reviewer #3

We are very indebted to you for your comments and helpful suggestions, which we have dealt with as follows:

1)      P.10/393: “or splicing regulatory elements (ESEs, ISEs, ESSs or ISS” -> or splicing regulatory elements BINDING SITES (ESEs, ISEs, ESSs or ISS). The text has been modified as suggested.

2)      Title: “3.3. Alternative splicing caused by changes (misbalance?) in splicing factors” – May it be made more specific on “changes”? I suggest: “3.3. Alternative splicing caused by disruption in splicing factors homeostasis/balance/expression level”, just it should be what you mean by that. To avoid misunderstandings, we have modified the text in the sense you suggested.

3)      Fig. 4 A: It would be worth assessing if the AS types distribution is shifted from the whole genome stats. You may find it in an appropriate paper. It looks like it does not. You were right; our data on driver genes do not follow the pattern found in the whole genome and we have added a new reference to emphasize these differences.

4)      The authors rightly state that “Changes in the level of splicing factors is a cause of AS alteration in driver genes EVEN (add) more frequent than mutation”. It’d be good to elaborate on the possible causes of levels alteration. I’d like to state the SFs homeostasis may be disrupted by the environmental stress. For example, ischemic insult shuts neuron expression rate close to negligent level, and the SFs expression rate goes down not proportionate, e.g. homeostasis will be disrupted. So, many types of stresses may lead to malignancy. Although we mentioned the effects of hypoxia on the level of splicing factors, we have now included a more general paragraph, in the sense you suggested, at the beginning of section 3.3.

5)      English maybe worth some polishing to a reasonable extent. The text has been modified in several places.

Reviewer 4 Report

Comments and Suggestions for Authors

The manuscript by Francisco Gimeno-Valiente and co-authors is a systematic review of the importance of alternative splicing and mutations in the splicing machinery. Alternative splicing is a natural mechanism to increase the diversity of gene functions. The oncogenic selection could select the splice variants that are important for tumorigenesis. Currently, the field of alternative splicing is overshadowed by other popular to study mechanisms of oncogenesis. Therefore, the present manuscript is of special interest to the Cancers journal and to the readers in the field of cancer research.

Minor comments.

1) TP53 is an important tumor suppressor gene. It has a huge variability of splice forms, but two paragraphs devoted to the TP53 gene in the manuscript do not properly characterize the involvement of different splicing forms in oncogenesis. The work of Sir David Lane, who has spent many years characterizing TP53 isoforms, should be studied and cited. The chapter should be expanded.

2) The information in Table 1 should be carefully re-evaluated. The fusion gene (t(8;21)(q22;q22) translocation) RUNX1/RUNX1T1 is a known transcriptional repressor. This information should be reflected in Table 1.

Author Response

Reviewer #4

Thank you very much for your useful comments, which have been dealt with as indicated below.

1) TP53 is an important tumor suppressor gene. It has a huge variability of splice forms, but two paragraphs devoted to the TP53 gene in the manuscript do not properly characterize the involvement of different splicing forms in oncogenesis. The work of Sir David Lane, who has spent many years characterizing TP53 isoforms, should be studied and cited. The chapter should be expanded. Following your suggestion, the text dealing with TP53 has been modified and expanded.

2) The information in Table 1 should be carefully re-evaluated. The fusion gene (t(8;21)(q22;q22) translocation) RUNX1/RUNX1T1 is a known transcriptional repressor. This information should be reflected in Table 1. Thank you for calling our attention to this role of RUNX1/RUNX1T1, which we had overlooked.